# Alibis of Exclusion: The role of ethnic economies in the differentiated inclusion of refugees in Berlin

Jagat Sohail

Princeton University

jsohail@princeton.edu

Friday, 25th November, 2022

**Abstract**

Neoliberal transformations in the field of asylum in Germany have, since 2016, placed an emphasis on labour market participation as the primary way through which refugees can establish long-term residence claims. Yet,  while new arrivals are increasingly expected to rapidly integrate into this market, they are often armed with differential and precarious legal statuses which overwhelmingly determine the spheres of economic activity refugees can enter.  This is especially true for the ever increasing number of rejected asylum seekers who are temporarily "tolerated", for whom the only path to residence requires that they display their value as economic subjects to the German state. Based on ethnographic fieldwork conducted in Berlin between 2017 and 2020, I argue that, armed with little other than their cultural identities and networks, many refugees heavily relied on co-ethnic relationships to facilitate their participation in the labour market. The result is a displacement of the management of new diversity to ethnic economies, a move that disavows the marginalising consequences of the neoliberal transformations refugees are subject to. By masking a relationship between recently arrived migrants and the German economy and society as a relationship between migrants themselves, I argue that these practices work as *alibis of exclusion*.

**Introduction: A Pakistani, A Syrian, and an Egyptian Walk into a Camp…**

In the summer of 2015 Khaled, a 24 year old Egyptian man, saw an arrest warrant issued in his name. The Egyptian regime had identified him as one among many political dissidents that had taken to the streets during the Arab uprisings that began in 2011. The warrant made him confront the blunt reality of his prospects in Egypt. Recognising the impossible constraints placed on him, he paid his way into a boat leaving from Alexandria that would take him to Greece. A few months prior, Omar, an 18 year old Syrian man, had been living by himself in Damascus, when an encounter with the regime set his plans in motion. Omar grew up in Dara'a, often referred to as the cradle of the Syrian revolution. His home, marked on his national identity card, made him particularly vulnerable at the unavoidable military checkpoints scattered across the city and country. Almost inevitably, he mis-stepped, angering someone with military connections. The swift retaliation he faced became his trigger to make his way north, and in quite spectacular fashion, he managed to take a flight to Turkey, finally paying a smuggler to get on a boat to Europe. Meanwhile, Haider, a Pakistani man in his 30s, was planning out a route that would allow him and his family safe passage across central Asia and into Europe. As a Shia Muslim, he faced everyday violence and discrimination in his village in Punjab, but things had escalated after his marriage to Abida. For months he received death threats from Abida's in-laws from a previous marriage. When some violent encounters proved the threats credible, Haider, a pregnant Abida, and their four year-old son got onto a bus that would take them to Iran. From there, they walked, hitched rides on buses, hid in trucks carrying cargo, and walked some more until they finally reached Turkey, where they too got into a boat that would take them onwards to the promised security of Europe.

Their journeys brought them to Germany, and by 2016, all three men were living in the same camp in the East of Berlin, where I would meet them for the first time a year later when I began conducting ethnographic fieldwork with refugees in the city. For their part, all three men were keen to leave behind their dependence on social workers, their pre-fabricated polymer shelters, and an environment of supervision, and begin their new lives in Berlin in earnest.

Work, they believed, would be their ticket out of the exceptional liminality of the camp, and into the diverse heterogeneity of the cosmopolis. Through labour, they could integrate, demonstrate their worth, shed their spatially segregated identities as refugees, regain their independence and, perhaps most crucially, their dignity. However, over the next few years, the camp would prove to have a gravity that was harder to resist than any of them had anticipated. Omar was the first to find an apartment outside, but, for his first job, he found himself back at the camp, working as a translator and social worker. Khaled was next and, having obtained a training certificate in private security, he too returned to the camp, stationed now as a security guard. Haider, despite working various odd jobs,

never quite made the transition out, and indeed, when he did finally leave, it was only because he and his family were relocated to another camp further away from the city centre.

The three men had arrived in Berlin as asylum seekers, structurally united by their need for safety, their perilous journeys to Europe, their position as strangers to German society, and a shared precarity in resources. Yet, a process of differentiation seemed to have quickly taken hold where the three men, soon related to one another as resident, guard, and social worker. How did this happen? What logic holds together this system of differentiation that sorts newly arrived asylum seekers in Berlin? How does it relate to work, and what explains the centripetal force exerted by the "camp"? This paper attempts to unpack this gravitational field that pulls newcomers into orbit, while simultaneously working to produce hierarchies of difference from within.

The insights presented here are based on over three years of ethnographic field research I conducted with asylum seekers that arrived in Berlin in and around 2015/16. In the years since, while some have gone on to obtain protection statuses and permanent residence, others have had their pleas rejected, wait on appeal decisions, or been equipped with intermediate bureaucratic categories that place them somewhere in between a continuum of the legal statuses that lie in between rejection and asylum. In this paper, reflecting the self-identification of my interlocutors, as well as their defining entanglements with the bureaucratic field of asylum, I use the term 'refugee' to refer to those of my interlocutors that came to Berlin seeking asylum, irrespective of the legal statuses they were then granted by the German state. Where legal or temporal specificity is important, I use the terms "asylum seeker", "rejected asylum seeker", "tolerated foreigner", "refugee with asylum status" etc.

I arrived in the city hoping to examine the aftermath of Germany's, at the time quite exceptional and exceptionally publicised, culture of welcome and hospitality. The period, referred to as *Willkommenskultur*, marked what many at the time believed presented an alternative imagination of how state and civil society could engage with newcomers. At the beginning of my research, I spent three months volunteering as a translator at a refugee camp in the east of the city, where I made many of my initial contacts. Over the course of the next three years, though I returned to the camp to visit some interlocutors that continued to live there, as most moved out, my fieldwork did too, and I spent hundreds of hours at various spaces of significance for refugee life in the city. These included a refugee camp in the east of the city, a refugee-led organisation in a diverse neighbourhood in the north, community and neighbourhood centres dealing with newcomers, and various cafes, parks and public hang-out spots. In total, I collected notes on informal conversations and encounters with over 60 refugees, though the bulk of my ethnographic work centred around 12 refugees distributed across three core friend groups of refugees, one Punjabi-speaking and two Arabic-speaking. In addition to

refugees themselves, I interviewed and spoke to various social workers, volunteers, and political activists engaged with refugee-related work and activities.

During my fieldwork, I encountered, and spent time with, refugees from various national cohorts, though, for linguistic reasons, those from Arabic speaking regions, as well as Punjabi or Urdu speakers from Pakistan constituted the majority of the people I spoke to. The diversity of identities in my fieldsite gradually made me appreciate the unevenness of experiences built into encounters of hospitality, and practices and trajectories of incorporation. In other senses, however, my interlocutors were quite uniform. My position as a young man in these spaces meant that I often had very limited access to women asylum seekers. As a result, my questions and insights were increasingly centred around engaging with the experience of male refugees. Further, my position as an Indian ethnographer in Germany was an odd one. As an outsider and newcomer to German society myself, my interlocutors were often already much more familiar with the city and German society than me. In many senses, I was introduced to the vagaries of Berlin life and norms through my refugee friends and interlocutors. The result, I believe, is a perspective on foreigner incorporation that focuses more acutely on the priorities, pressures and perspectives of refugees themselves.

When I began, I did not intend to study the economic lives of my interlocutors specifically, but rather the broader dynamics of foreigner incorporation they were involved in, especially as initiated by interpersonal encounters in 2015/16. However, it quickly became clear to me that the field of labour was both unavoidable, and central to the nature of their engagements as and with members of Berlin's social world. I began to realize that many of my interlocutors seemed to find themselves "stuck" in specific, segmented zones of economic activity, that seemed to be determined by their passports. Across Europe and beyond, scholars of refugee and migration studies have identified processes of "differential inclusion" (Könönen, 2018; Mezzadra & Neilson, 2013), "precarious integration" (Maroufi, 2017), "precarious inclusion" (Rytter & Ghandchi, 2020), or "integration through disintegration" (Hinger, 2020)to indicate processes whereby the selective incorporation of foreigners into the body politic produces new logics of ethnic segmentation and boundary formation, producing, what Nicholas DeGenova (2013)refers to as the "obscene of inclusion", a " process of inclusion through exclusion" where migrants are incorporated primarily as subordinated labour.

Yet, throughout my fieldwork, while I was frequently witness to precarious and exploitative working conditions, what I found more challenging to articulate, was that refugees did not experience these relationships only with the state, nor only with German economic institutions and bosses, as was often the case with the Guest Workers of a previous generation. Instead, for many of my interlocutors, their immediate exploitative relationships tended to be with members from their own linguistic, national or ethnic communities. On the face of it, it should surprise no one that, faced with limited and

differential access to formal paths to employment, asylum seekers turn to their social networks to look for jobs. Yet it soon became clear to me that the nature of the relationships thus forged, rather than expressions of community solidarity, were often experienced as acutely exploitative and poisonous, and in ways that were significantly exacerbated by policy changes in the field of asylum. Many of my interlocutors found themselves engaging in severely underpaid and precarious labour for an older generation of migrants with whom they shared some kind of cultural common ground. In many cases, they were then put to work  policing other refugees and migrants that they shared either legal or cultural commonalities with. Yet other interlocutors looked for ways to profit from an aesthetic industry that functioned through the precise commodification and consumption of refugee experience and identity. This paper attempts to conceptualise how this became possible, and will make the argument that, under neoliberalism, these "ethnic economies" become both the engine and alibi of the German state's attempt to produce value from recently arrived asylum seekers and refugees. This is made possible by two interrelated moves. The first *displaces* the task of the management of new differences to co-ethnic spaces. The second, involves an *aesthetic shift* that parallels an increasing neoliberalization of the field of asylum, working to *disavow* the marginalising consequences of these policies. By masking a relationship between recently arrived migrants and the German economy and society as a relationship between migrants themselves, I argue that these practices work as *alibis of exclusion*. Furthermore, understanding this shift is essential, if we are to make sense of the sorting and stratification of  asylum seekers that arrived in Berlin in the years following 2015. In doing so, I hope to shed light on the internal dynamics of processes of differential, and differentiated, inclusion.

The central questions I will address are the following: *How are migrants, asylum seekers and refugees themselves involved in the maintenance and legitimation of differential inclusion? How does state policy produce, and determine the nature of, co-ethnic labour relations? How can we understand the material and Ideological role played by co-ethnic labour relations under neo-liberal transformations?* In this paper, I will demonstrate the way my interlocutors often seem to find themselves in these new ethnically sealed zones of economic activity - returning, metaphorically, to the "camp" - despite their attempts to escape into other spheres of labour. However, despite appearances, these zones or enclaves, I argue, are not just the straightforward result of refugees instrumentalizing their social capital to establish new economic lives, but the structural consequence of neo-liberal policies on the one hand, and the aesthetic and moral politics of Berlin's cosmopolitan culture on the other. This shift, in turn, is made possible by an ever increasing proliferation of legal statuses in Germany that fracture and differentiate the field of asylum today.

The article begins with an analysis of the changes introduced in asylum policy and management in the years after 2015. I show how state-directed pressure to rapidly enter the labour market combined with a bureaucratic system of asylum management that sorted asylum seekers into an increasingly stratified

set of legal statuses that overwhelmingly determine the nature and location of their labour practices. I then show how the concept of the *Bleibeperspective* (staying prospects) introduces a moral economy that differentiates "real"and "bogus"asylum seekers through the lens of deservingness, a lens that has less to do with the biographical narratives of asylum seekers than with their membership within certain national cohorts. Through Haider's experience, I then show how precarious legal statuses are expressed in the "ethnic economy", and the poisonous effect that they have on co-ethnic relationships between recently arrived, and longer settled migrants. With an analysis of Khaled's experience in the private security industry, I then show how neoliberal ethnic economies can also become aesthetic and moral structures that mask and disavow the work of disciplinary practices and social control. Finally, I introduce Omar to show how refugees can themselves become involved in the production and reproduction of these new hierarchies and moral economies, by internalising neoliberal notions of entrepreneurial selfhood.

 I chose to tell the stories of Omar Khaled and Haider because of how their experiences represented three distinct trajectories of  labour incorporation and its relationship to co-ethnicity. Each had, as a result of their search for employment, unique relationships with other refugees, as well as with longer settled migrants of their own linguistic, national, or "ethnic" backgrounds. I began to think of their stories as representative - not necessarily in a statistical sense - of the dominant modes of engagement with the labour market for my interlocutors. Such a choice prioritises a kind of analytical specificity over complexity, eschewing stories that may not fit so clearly into the analysis that follows. Yet by doing so, I hope to articulate a possible way to think of the structural forces my interlocutors found themselves engaging with. Together, the stories not only address the differentiation of recently arrived refugees into socio-economic hierarchies, but also how "co-ethnicity" is itself produced. As I argue, ethnic boundaries exist not because of sui-generis proximity between people from similar cultures, but because of the way these boundaries produce value, both in the economy, as well as in the task of the management and control of new diversity. I follow scholars working with theories of "Racial Capitalism" to elucidate how these material and ideological shifts mask the consequences of neo-liberal restructurings in the field of asylum and migration.

**Asylum and the Neoliberal Turn**

Beginning in 2014, Germany introduced significant policy reforms to allow asylum seekers into the labour market. This gained crucial momentum with the Integration Act of 2016, and the new motto of labour policy for asylum seekers, Fördern und Fordern (Encourage and Demand). Under the new regime, asylum seekers would be provided paths of entry into the labour market, and their participation as workers would become the key indicator used to evaluate their residence claims, and determine the extent of their access to welfare resources through a graded system of sanctions. This

new approach did two things. First, it converted the question of integration into a primarily economic one of participation in the labour market. Second, it shifted the onus of this economic integration onto asylum seekers, especially those whose initial pleas had been rejected, who would now have to prove their worth if they were to continue to stay in the country. Referring to the new labour reforms, the Bundesregierung website says, "Refugees who have good prospects of staying will receive offers from the state through the integration act early on. However, they are obligated to strive for integration themselves" (Bundesregierung 2016).

The reforms introduced a "paradigmatic shift guided by labour market considerations"(Scherschel, 2016, p. 246) in Germany's approach to Asylum. Scholars of political economy have located this shift within a post-fordist context that seeks to economise relations between the citizen and state (Jessop, 2016; Maroufi, 2017)), within the transition from welfare to workfare (Eick et al., 2003). This began an economising of the field of asylum, indicating a serious blurring of the boundaries between migration and asylum. While the impossibility of separating these fields has been a central feature of the study of forced migration (Bartram, 2015; Crawley & Skleparis, 2018), this shift in state and public discourse is significant. Indeed until very recently, Germany had distinct migration and asylum policies. Where the former was explicitly part of the country's economic policy, asylum seekers faced serious hurdles in accessing the labour market, a policy meant to dissuade people thinking of applying for asylum in Germany (Hinger, 2020, p. 19). This is particularly relevant considering that the number of asylum seekers denied asylum, but temporarily allowed to stay in the country has steadily been growing. Stuck in bureaucratic limbo, hundreds of thousands of people who could neither leave, nor really stay, presented an untapped resource that the state began to evaluate economically.

Indeed most newcomers applying for asylum in Germany, that cannot prove a select few identities, have their asylum pleas rejected. Rejected asylum seekers may appeal their decision. During this period they may not be deported and, since the procedure invariably takes years to be processed, it is in this time that rejected asylum seekers begin to look for other ways to earn their protection and gain permanent residence. As of 2021, the number of rejected asylum seekers living in the country was as high as 292,672. Of these, 242,029 held a "*Duldung*"[i] (Deutscher Bundestag, 2022) - a temporary status of "toleration" requiring weekly or monthly renewal that essentially implies that their deportation has been temporarily suspended . For reference, this makes them almost as large a cohort as the 317,835  refugees living in Germany with permanent residence permits (Statistisches Bundesamt, 2022).

Under the new reforms introduced in 2016, those with a *Duldung* could make claims to permanent residency by retraining and/or entering the labour market. By enrolling into an "Ausbildung" (vocational training course),  tolerated foreigners could gain access to an "Ausbildungsduldung",

which extends the validity of their temporary permit to cover the duration of the course and two years of employment after. This "3+2" rule introduced in August 2016, was formulated explicitly to take advantage of the vast numbers of such asylum seekers, a move that made practical sense given the labour shortages in skilled work in the lower segments of the economy. Importantly, this indicates the clear incorporation of labour market integration as one of the primary ways in which residence claims are evaluated, particularly in regards to what the state terms "gut integrierte Geduldete" (well integrated tolerated foreigners). As Drangsland argues (2020, p. 1129), "[t]he *Ausbildungsduldung* thereby appears as a bio-political mechanism for filtering migrants and for the differential investment in their lives, conditional on their ability to become a skilled worker."

It is important to note that, for years, groups like Pro Asyl, the major German NGO in the field of asylum, had actively campaigned for the right to work, both for refugees that received asylum, as well as rejected asylum seekers that ended up with uncertain legal statuses, often waiting in limbo for years as they appealed their asylum decisions. That those with a *Duldung* could work, and through it finally establish themselves as permanent residents is a major reprieve for many who have lived with the status in a kind of "permanent temporariness", that, in some cases, has lasted for decades (Tize, 2021). Yet, while many hailed the reforms as an essential, progressive step forward, scholars have emphasised how these changes were brought about, less by asylum seekers' own initiatives, and more by the lobbying activities, and commercial interests of German industries and groups such as the BDI, the *Bundesverband der Deutschen Industrie* (see Maroufi, 2017), and the BDA, the *Bundesvereinigung der Deutschen Arbeitgeberverbände* (see Drangsland, 2020). For years, industry lobbyists had emphasised the need for new sources of cheap, skilled labour, insisting that asylum seekers presented a potential resource that the German state needed to capitalise on. This distinctly neo-liberal turn is perhaps best represented by the central involvement of the global consultancy firm Mckinsey, in the delineation of the 2016 Integration act (Pichl, 2020). Indeed, scholars have argued (Drangsland, 2020; Schultz, 2019) that it is precisely the need for new sources of labour that underpinned Germany's decision to open its borders in 2015, suggesting that demographic projections of an ageing and shrinking population played a central role in framing incoming asylum seekers as an opportunity, and as unrealized potential labour. Rather than serving as an unqualified right for those who had lived with this status for years, these policies post 2015 were framed to allow the state to selectively recruit from the pool of tolerated foreigners depending on the specific needs of the German labour market. Thus, as Theresa Schütze (2022, p. 16) argues, "neoliberalization has not improved the situation for all persons with Duldung but has driven differentiation within the group and a rationale of 'cherry picking' by the state".

Thus, for most asylum seekers that received uncertain legal documentation – over 50% of those that applied between 2015 and 2021 did not obtain refugee status - work has been experienced less as a

right, and a pathway through which to actively participate in German society, than as an expectation that must be met at the cost of serious sanctions. For "tolerated foreigners", these sanctions range from a reduced access to welfare state support, to the looming threat of deportation. The incentive structures thus set up produced what scholars have referred to as a "work for safety" (Wyss & Fischer, 2022) logic, whereby the productivity of asylum seekers, and their ability and willingness to retrain according to the requirements of the German economy have increasingly begun to precede and replace protection and security as the moral basis for residence in Germany.  Indeed, for many, the current German solution produces a kind of extended and persistent state of everyday "deportability" (De Genova, 2013) that functions as the tangible atmosphere under which asylum seekers scramble to find work in Berlin.

Furthermore, it is important to note that this sanctions/incentive structure is not just experienced by rejected asylum seekers. Among those that are afforded some status of protection, almost half do not receive asylum (Statistisches Bundesamt, 2022). In 2021, while 21.4 % of total applications resulted in asylum status, 17.5% received other protection statuses that do not come with the full benefits of asylum , and so far, in 2022, the number of those without asylum has been lower than those with other protection statuses (BAMF, 2022, p. 11). Of note here is  "subsidiary protection" (15.3% in 2021), a status in between rejection and asylum that comes with a one year residence permit, periodically renewable for two years, as long as it is determined that the situation in the host country has not improved. Those with this status may apply for permanent residency after 5 years, but, among other conditions,  in order to do so they have to have paid into social security for 60 months, and been independent of state welfare during this period. Unlike those on a *Duldung*, those with this status have unrestricted access to the labour market. However, the likelihood of obtaining stable and long-term employment with temporary residence permits proves to be a significant challenge, and this is without mentioning the various other barriers that prevent displaced persons from rapidly entering new labour markets such as language, the non-recognition of credentials and the time needed to recuperate from trauma and loss. Finally those that receive either refugee status or constitutional asylum (both are functionally the same), receive permits for 3 years, and they too need to work in order to establish permanent residence in Germany, though the criteria in this case are much more forgiving, and they are allowed to do so as early as three years after their initial asylum plea. Yet, while the status does indeed make a significant difference, even for those who obtain full refugee status, working often implies extensive retraining, and/or a significant loss of employment status.

In other words, despite (or maybe because of) the state's ongoing attempts to make the most of its refugee population, the pressure is always downwards - first towards the lower segments of the formal labour market, and finally towards the informal and illegal margins of economic activity.  "The field of asylum", as Benjamin Etzold (2017, p. 99) argues, "has been divided into further subfields, in

which differently categorised groups of refugees are differentially positioned… It has enhanced the chances for rapid labour market integration for some – they can capitalise on asylum – but radically reduced the chances for others." The result is a gradation through which refugees are differentially equipped with bureaucratic categories and resources as they enter the labour market, as they seek to earn their safety in Germany.

**Politics of Deservingness and the *Bleibeperspektive* (Staying Prospects)**

While some may be tempted to argue that this bureaucratic differentiation of the field of asylum only reflects a natural hierarchy based on the validity and urgency of asylum claims, it is important to address the principle through which asylum seekers are sorted into these categories. Asylum cases are overwhelmingly linked not to personal history, but national identity. This was the direct result of an administrative innovation introduced by the Federal Office for Migration and Refugees (BAMF) whereby asylum cases were sorted into clusters depending on the applicant's country of origin. This is when the notion of "*Bleibeperspektive*" (staying prospects) became essential. The *Bleibeperspektive* created a clear hierarchy of deserving-ness for asylum seekers that was based on previous rates of acceptance for a national group, rather than targeted to individual histories and personal accounts.

Through this approach, applications from Cluster A (Syria, Eritrea, Iran, Iraq, Somalia), would be fast tracked for asylum. Those from Cluster B, however, were considered to be from 'safe countries of origin' meaning that the likelihood of their asylum pleas being rejected was extremely high. While the cluster system was officially discontinued in 2017, as Anne-Kathrin Will (2018, p. 178) argues, citizenship continues to be the decisive factor in the processing of asylum applications: "an asylum seeker's passport still determines whether he/she will be steered toward the fasttrack to integration (with more opportunities) or toward the fast-track to return (with more pressure)." Indeed, in many senses, the "*Bleibeperspektive*" - a statistically produced tautology - almost exclusively mediates the "legal division between putative 'genuine' and 'bogus' through the notion of strong or weak 'likelihood of staying'" (Hinger, 2020, p. 23).

Second, increasingly, scholars observe a "trend to grant refugees a temporary residence permit—and thus a precarious legal status—instead of long-term legal protection" (Wyss & Fischer, 2022, p. 630; see also O'Sullivan, 2019). In other words, the increasing proliferation and multiplication of uncertain and precarious documentation is increasingly emerging as an institutionalised response to the management of asylum seekers, and can no longer be thought of as the constitutive outside to the category of asylum. The *Duldung,* its various forms, and other in-between legal statuses, cannot be understood as a mere rejection of the status of refugee-ness. Nor does it have much to do, at all, with an attempt to evaluate the credibility of an applicant's claim for asylum. It is, instead, becoming one of

the various legal categories through which asylum seekers seek protection that is denied to them through the legal category of asylum.[ii]

A bordering practice thus emerges that sorts asylum seekers into different segments of the formal and informal economy on the basis of national identity. As Mezzadra and Neilson have argued, this multiplication of legal statuses is closely linked to the multiplication of labour. Bordering, through spatial, temporal and bureaucratic processes, is a way to make "ungovernable flows" governable (2013, p. 149). The goal, scholars contend, is "synchronizing migrants' mobility with the needs of labor markets" (Drangsland, 2020, p. 1130). In other words, it is important to recognize how these statuses, rather than lying 'outside' of citizenship or residence, work instead as a "handicap within a continuum of probationary citizenship" (Chauvin & Garcés-Mascareñas, 2012, p. 243).

The relationship of these policies to the segmentation of economic activity for refugees quickly becomes apparent. Those most unlikely to receive favourable documentation are often pushed into and outside the formal margins of the economy. These are imaginably the most vulnerable, and subject to the unfettered exploitation of the lowest informal and illegal rungs of the economy. A segment above are those that enrol in formal apprenticeships/ training programs[iii]. This comprises people belonging to a range of countries who bring enough forms of capital with them to make their entry into this segment possible. Notable here are Afghan refugees, who, since 2016, have almost as a rule existed a tier just below Syrians and Eritreans (Vo, 2016; Wyss & Fischer, 2022). The final tier of this hierarchy is constituted by those that have access to the full benefits of asylum though, as I will argue, they too are not immune from the effects of the neoliberal moral politics that defines deservingness through productive participation in the German economy.

Under the current regime, though the state expects asylum seekers and refugees to prove their value, they are denied the "legal capital" (Etzold, 2017, p. 84) that would allow them to do so. Indeed, somewhat paradoxically, deservingness seems to simultaneously become "both a civic obligation and a civic privilege duty"(Chauvin & Garcés-Mascareñas, 2014, p. 423). The result is that many cannot possibly produce value in the German economy, without accessing their social networks.

**Haider, the Tolerated Foreigner**

When I first met him, Haider was on a single-minded mission to find employment. Echoing the entrepreneurial subjectivities expected by German state discourse, Haider insisted that he needed, above all else, to find stable employment, and begin paying taxes as soon as he could. "Just by giving us space to be safe, they have done more than enough for us", he told me early on, "I want to pay them back." I later realised that a more existential theme underpinned his desperation to find work.

Haider and Abida's asylum pleas had been rejected. Though they planned to appeal the decision, this merely bought them some time. The chances of overturning the decision were extremely remote, and for now Haider was classified as a "tolerated foreigner".

During my time as a volunteer in the camp, social workers asked me to talk to Haider about enrolling into a vocational training course, with which he could get an *Ausbildungsduldung*. They believed this to be the only hope in his case and, as the only Punjabi and Urdu speaker in the camp, they hoped I could convince him that this was in his best interest . Though I did speak to him, I quickly realised that for Haider, and many others like him, the *Ausbildungsduldung* did not offer a realistic path forward. Retraining in a new language and new field, at his age, was a daunting prospect, particularly given the time constraints within which he would have to achieve this. Further, dropping out of the Ausbildung at any point, which is not an infrequent occurrence given the language and educational barriers (Janczyk, 2018), would leave him with only 6 months to find something else, failing which he would be deported, and these were all risks that Haider told me he wasn't willing to take. Later, in January of 2020, the German government introduced an additional variation of the *Duldung* - the *Beschäftigungsduldung* - which would allow tolerated foreigners to stay in the country provided they can prove at least 18 months of continuous employment during which they have paid into social security, along with not having relied on any welfare for at least 12 months. This too was a tall ask. In Pakistan, Haider had worked in a textile factory, and his skills, in an increasingly service-oriented city economy, were not easily transferable.[iv]

Nonetheless, recognizing the underlying principle at work, Haider was adamant that, if he worked, the German state would start seeing his family's worth, giving them the push that they needed to become legitimate members in their adopted home. However, Haider spoke almost no German. The countdown the appeal process started was slowly ticking down to the eventuality of deportation - Pakistan was a designated safe country, and the German and Pakistani governments had recently signed an agreement which would help facilitate deportations of Pakistan nationals. Taking valuable time off to study was not an option. In addition to this, armed with only a Duldung, the formal economy was almost entirely out of reach. Reckoning he needed to start somewhere, Haider would set out daily in the hope of finding some kind of informal arrangement. When I first met him, work had remained elusive, though this hadn't always been the case.

The only work he got previously, through his South Asian contacts, was back-breaking and humiliating. At one point, he worked, off the books, as a cleaner in a hotel run by a Pakistani man for 2 euros an hour - one fourth the legal minimum wage. The man used every opportunity to berate him for not working quickly enough, even as Haider developed serious back problems as a result of the job. "I had to get his permission to breathe", he told me. At various times he had been at the receiving

end of caste and class based slurs and insults, the constant indignity of which had taken a serious toll on his mental well-being. One employer even refused to call him by his name, referring to him only as "paindu", a derogatory term which literally means "from the village" and has strong casteist undertones. He had decided against working for South Asians again. Instead, he hoped to find a willing Turkish employer - he'd heard that they were more likely to be fair, though his attempts remained largely unsuccessful. It is what he summarised that stayed in my mind - "Sachch ta aa hi hai - sabton zyaadan, apne hi apno nu lootdene" (This is the real truth - our own people loot us the most). As I hope to show, with the increasing proliferation of legal statuses and labour-centred incentives to residency, what we are seeing is an intensification of this structure of migrant labour exploitation, where migrants are put to work by other migrants in ways that are particularly hard to see and define.

Scholars working on migration have identified the ways in which economic activity can become organised within "ethnic economies" or "ethnic enclave economies" in cases where these labour markets also take on geo-spatial features within cities. Yet while much of this literature points to the importance of co-ethnic networks in the absorption of new migrant labour, scholars have been somewhat hesitant to point out the role played by state policy in producing the ethnic economy, and how and when co-ethnicity becomes a source of exploitation, rather than solidarity. To an extent this has been because the backdrop of the ethnic economy is constructed as the state's absence, not its presence. As such, the mobilisation of co-ethnicity becomes reactionary, a "self-defense of immigrants and ethnic minorities who confront exclusion or disadvantage in labor markets" (Light, 2005, p. 6520). The tendency, therefore, has been to frame these intra-ethnic relationships as the play of social capital amongst refugee communities (Drever & Hoffmeister, 2008). That, in his position, Haider would rely on contacts within his own communities to find work is not surprising, but the framing of this as migrants exercising instrumental agency through social capital tends to not question the conditions under which these social networks become significant social capital in the first place. It tends to obfuscate how neoliberal economic restructuring, combined with the withdrawal of the welfare state paradigm, turns cultural knowledge and relationships into specific forms of capital that sharpen the inequalities of migrant labour, turning "social capital" into "networks of exploitation" (Cranford, 2005).

As it turned out Haider's search for work outside of the social space that he defined as "our own people" would prove unsuccessful. Some explicitly cited his legal status, but for many, it came down to the absence of a common language. Unable and unwilling to forgo an income and put his ever-shrinking time and resources into learning German, he reluctantly returned to a Pakistani employer, this time working in an electronics store. A few months after our earlier conversation about his search for work, I sat with him and two other younger Pakistani men who shared the same legal status. While they too were complaining about the harsh and unforgiving nature of working for their employers

(both worked in an Asian supermarket run by a Pakistani man), Haider interjected and said simply, "Duldung de naal aa hi kam mildaa ai" (with a *Duldung* this is the only work you can get). Haider's words echoed a well established fact in the study of labour migration, namely that a "compromised socio-legal status… is central to… unfree recruitment into forced labour" (Lewis et al., 2015, p. 89), yet it also underlines another important aspect, namely the effect these statuses have on migrant's "social worlds" (Sigona, 2012).

Rocio Rosales (2014) has shown how, in the absence of access to stable documentation, Mexican immigrant fruit vendors in California became roped into "cycles of exploitation", with escape often proving only possible through radical departures back to Mexico. This perspective is vital. For Haider, who occupied the bottom rung of the bureaucratic hierarchy, his deportability and precarious legal status significantly affected not just his relationship with the German state and society at large, but also with other members of the Pakistani and South Asian communities in Berlin. The image of a top-down structure of marginalisation against which migrant communities leverage network ties prevents us from taking account of the unevenness of migrant experience. As Neha Vora and Natalie Koch argue, in the context of South Asian labour migrations to the Gulf, such an image "erases the important role of non-nationals themselves in the processes of migrant governance and labour exploitation" (2015, p. 543).

 Yet this is an unevenness that, while being produced by state policy, simultaneously exceeds it, introducing elements of cultural reason and difference that come from the communities themselves. Haider's South Asian employers were able to draw on their reservoirs of social and cultural capital to read his situation in a way that he no doubt hoped employers from other ethnic groups might not be able. Their incessant use of casteist slurs and referring to him as from the "village" were not just meant to humiliate, but to also discipline and control through the performative reproduction of difference. In other words, while the "co-ethnic" analytic is crucial, it is equally important to note that movement into such spaces isn't marked by a transition into homogeneity. Quite the contrary, it is precisely the existence of difference and hierarchy of various kinds - caste, sect, social class, dialect, rural/urban etc. - that become operationalized in these spaces. It is the mark of a difference, however, that is more proximate, nuanced, and culturally informed, than the broader, perhaps even more radical, difference between 'newcomers' and 'Germans'. Yet it is precisely the proximity of this difference that allows them to become operationalized in the German context.

Intra-ethnic boundaries and hierarchies thus perform crucial functions of distinction that play important roles in both identity formation (Charsley & Bolognani, 2017) as well as, as Biao Xiang (2012) argues, in regulating and monitoring diversity through new practices of labour. In his excellent work on the management of migrant labour in China, Xiang shows how the state relies on private

intermediaries to "turn flesh and blood migrants into "paper migrants," to transform unpredictable individual mobility into legible, aggregate flows, and to hold agents as scapegoats if needed" (2012, p. 51).  This last point is significant, and perhaps the least underappreciated aspect of the role of ethnic economies in neo-liberal restructurings - the way they work as scapegoats or alibis for the marginalising consequences of state policy. Intermediaries, whether formally recognized, as in Xiang's work, or informally organised through the ethnic economy, mask the role played by the state in producing precarity. Indeed, this is reflected in the way Haider speaks of the German state as fundamentally good, but "our own people" as the ones that are responsible for his marginalisation, even as he recognizes the role played by his legal status in determining the nature of his labour position. To further articulate how this alibi takes shape, the next section follows the case of private security in Berlin, and the role played by refugees like Khaled in its functioning.

**Khaled, the Insecure Security Guard**

From the beginning, it was clear to everyone who knew him that Khaled was deeply driven. His German is excellent, and it's clear that he's been more successful at life in Berlin than many others in his position. After just a few months of knowing him, I had already heard about at least five different professions he fantasised about. So it seemed a little strange to me when, instead of pursuing any one of his various goals, he began working as a security guard. For a while, he was back once again, at the same camp that he had only just moved out of. Whenever I asked about the work, he was evasive, focusing instead on the fact that he was studying IT by himself while at work, or even that he was less a security guard, and more a social worker, helping residents of the camp with translation work. He regularly distanced himself from guard labour, insisting that it was merely a temporary pitstop on his way to something bigger. In 2020, Khaled's residence status was up for renewal. But the office in charge was inaccessible due to the Covid-19 lockdown and, as a result, he temporarily lost his work permit. Only after losing his job did he begin to open up about the bitterness he felt towards his job and employers. So much so that even after his paperwork was resolved, Khaled refused to return to guard labour, insisting that it was "bad work" and that he couldn't bear to spend his days intimidating people.

\*\*\*

A few months after he lost his job, Khaled and I sit together at a quiet spot at one of Berlin's many lakes on a warm day in September. Khaled tells me about a time last year when he worked for one of Berlin's largest security companies, run by a Lebanese-German man who was infamous among the refugee community. He was brought on board for a team responsible for security at a Christmas market. All of those employed were migrants, half of whom were Arabs. I ask Khaled why there

weren't any Germans. He tells me it was obvious, " because we're Arabs they feel safer because they think that we will identify the next Anis Amri[v]… That's our job, you see? it's to keep the Arabs out so everyone else can feel safe." Frowning, I ask him why so many Arab refugees continue to work in this industry if this was so obvious to everyone.

"Like what about those guards at SO36? " I'm referring to a time when Khaled, a few friends and I, were turned away from the gates of an iconic Berlin club on their "homo-oriental dance night" by two Arab security guards. This pattern - of brown and black men being the gate-keepers for traditionally white spaces has been an increasingly common part of everyday frustrations for my refugee friends in the city.  "Even those two guys you know, they probably didn't like their job". Khaled explains,  there are basically two major subcontractors in Berlin, both run by longer-settled migrants from the Middle-East,  that are likely to employ refugees. But they know exactly how desperate they are for the job, because it's linked to their residency status. So, they keep people on the edge and ensure that no one is able to claim benefits of full-time employees by offering part-time contracts that rotate every six months. With insecurity comes unquestioned obedience. The model works, Khaled continues, because they provide people like them who work for less than the Germans, and because they're so scared of losing the job, "we'll always do what we're told. So, if I'm told to keep an eye out for Arabs... then that's what I'll do. It's really shit work."

\*\*\*

Even for those that possess legal statuses above the *Duldung*, few jobs in the formal economy are easily accessible to newcomers in Germany. For young, male refugees like Khaled, one of the easiest paths to a job - one that requires the least amount of time, and the least amount of context specific knowledge (like German language skills) is that of "Sicherheit", or private security. At times it felt like every second young, male refugee I met was either working, or looking to find work, in this industry. The common explanation I received for this from my interlocutors was the ease of obtaining the training certificate.

A straightforward instrumental response by refugees to the range of options laid out in front of them is somewhat complicated when one realises that the security industry is one of the fastest growing industries in the country. The role of refugees in this growth complicates things. Loic Wacquant (1999, p. 220) suggests that the increasing incarceration of foreigners, "gives a precious and prescient indication of the type of society and state that Europe is in the process of building."  Wacquant popularised the idea that social control under neoliberal governance involved a re-emergence, in Western states, of penalization and policing as a method through which states manage an ethnoracial underclass in the pursuit of the logic of capital. While Germany hasn't seen quite the same growth in

its prison population as the United States, he suggests that "European societies endowed with a strong statist tradition are using the front end of the penal chain, the police, rather than the back end" (Wacquant et al., 2011, p. 206). In Germany, this front end has also undergone a serious transformation, with the dramatic rise of privatised security companies that, since the 1990s, have taken over much of the state's role in the management of urban populations, and have "come to be understood as the missing link between "civil society" and the state police" (Eick, 2006, p. 66).

This growth shows no signs of slowing down. In 2015, the industry recorded an annual turnover of 6.96 billion Euros. A year later, by the end of 2016, it accounted for a staggering 8.85 billion Euros (BDSW et al., 2021). The 27% single year rise coincided with Berlin's long summer of migration. This was certainly clear in the internal conversations in the industry. Harald Olschok, the general manager of the Federal Association of the Security Industry acknowledged that, "The protection of refugee shelters has led to this absolute boom" (BDSW, 2016). This trend continues to hold, and the presumed criminality and supposed threat posed by asylum seekers arriving to Germany's shores has no doubt been central to this dramatic growth.

The consequences of this are worth noting. For years the industry was a magnet for the far-right (Eick 2006). The lack of legislation, minimal training, and the ease of obtaining jobs in the industry have meant that this trend had serious effects on asylum reception centres. Cases of violence against refugees by guards, or even of refugees being forced into drug dealing or prostitution rackets by the people stationed to protect them made frequent media headlines (Deutsche Welle 2017). As Priska Komaromi (2016, p. 80)notes, "[n]ot only does the increased privatisation of asylum care actively put the lives of asylum seekers at risk, it also allows the state to absolve itself of responsibility and fails to ensure accountability and justice for the abuses that are committed against asylum seekers."

This logic, whereby security companies become alibis for state control and policing, forms the functional back-bone of these companies and their position in neoliberal restructurings of governance. "In praxis, rent-a-cop companies embody much of the power and privileges of the state, while bearing none of the responsibilities and limitations of democratic government." (Eick, 2006, p. 79). Yet, this alibi, it turns out, might still be incomplete. What might we make of the fact that increasingly, refugees seem to be enlisted side by side with neo-Nazis in the regulation of refugee mobility and access to urban space in Berlin? If the point is for security companies to work as alibis for state force, then this might indeed make sense. As people in the security industry agree, all the stories of unvetted staff attacking refugees created an "image problem" (BDSW, 2017), and hiring refugees might indeed turn out to be one of the solutions.

All of my interlocutors who worked as security guards were recruited by subcontractors who worked as middle-men for larger companies. Indeed, many of them were recruited directly from the camp where I volunteered, and often they worked in cohorts constituted almost exclusively of recently arrived refugees like themselves, with Arabic-speaking refugees forming the majority. While they wore the uniforms of the parent companies they worked for, their contracts and access to work were almost exclusively controlled by the middlemen who had recruited them. And though they had legal work contracts, a significant degree of informality nonetheless defined their labour experience. Most I spoke to were kept on limited and temporary contracts, but were often, illegally, made to work several additional hours off the book. This strategy of "organised informality" (Gooptu, 2013, p. 10) is central to understanding the functional role played by subcontractors who provide labour at lower costs while limiting legal liability for parent companies who conveniently look away from these practices. The middlemen themselves were longer-settled migrants from the Middle-East who could speak with camp residents in Arabic. In Khaled's words they were "al arab mithluna" (Arabs like us).

The phenomenon of migrant middle men working as intermediaries or "brokers" (Lindquist et al., 2012) between local companies and foreign labour is one that has been described in migration contexts around the globe (van den Broek et al., 2016; Xiang, 2012). In Germany too, in the past decade, researchers have shown how these logics operate in various spaces like the meat packing industry (Wagner & Hassel, 2016), construction work (Voivozeanu, 2019) and agriculture (Cosma et al., 2020). Yet not only were my interlocutors recruited by people "like them", many of them were also then put to work policing and surveilling others "like them".  For Khaled, this structure was what made guard labour unbearable. He frequently found himself in positions where he negotiated, on the one hand, with his own precarious working relationships with his bosses and the disciplining roles he was forced to take with people he identified with, on the other. A combination of his own desperation and the need for stable employment meant that he found himself commoditizing his cultural knowledge in the pursuit of a mode of control that is both effective and, crucially,  shielded from outrage.

It is precisely the aesthetic innovation that is crucial here. Getting refugees involved in the surveillance and disciplining of refugees allows companies to mask the policing of newcomers under the guise of "Kultursensible Arbeit" (culturally sensitive work). Using Migrant subcontractors that in turn hire asylum seekers and refugees, makes it possible to address how this policing "looks" without having to address the serious short-comings in the regulatory processes that might hold them accountable. Instead of having to materially invest in training and oversight, it becomes possible to push profits further with a highly compliant workforce who depend on these jobs for their existential safety, and the right to stay in Germany.

 Where Haider's case illustrates how, for those at the bottom of the material and moral hierarchy of the field of asylum, the ethnic economy becomes a proxy for practices of wage appropriation, Khaled's experience, from the middle of this hierarchy, adds biopolitical practices of disciplining and surveillance to the mix (Foucault, 1995). There is one final description that this paper now turns to, one that represents the segments of the refugee population that have received the full support of asylum, and helps to an extent, to display how refugees themselves become invested in reproducing and maintaining the hierarchies they find themselves in.

**Omar, the Middleman Entrepreneur**

Omar arrived from Syria in Berlin at the peak of Germany's *Wilkommenskultur* moment in 2015 when he was 19 years old. Already by 2017, Omar had obtained his C1 German proficiency - significantly faster than most others at the camp. Perhaps even more importantly, he quickly picked up the aesthetic language of the city. That Omar seemed to be particularly successful at transforming himself didn't go unnoticed among the group of friends he had made at the camp, many of whom often teased him of being too eager to become German, and how he seemed obsessed with appearing older and more independent than he actually was. Indeed, at the first opportunity he had, Omar moved out of the camp, even though this meant moving alone into an expensive flat in the city centre. Yet attempts at distancing himself from being identified as a young, dependent, Syrian refugee were more complicated. Omar's lack of training and spotty education meant that  he had practically no edge in the labour market, unless he was willing to commoditize those specific aspects of his own identity that would allow him to stand out. In other words, to economically integrate, the only realistic route for Omar was to monetize his position as a well-integrated outsider.

As a consequence, the first part time work he found involved working as a translator at the camp that he had been trying so hard to escape. While he found ways to distinguish himself from camp residents, the Syrian and refugee aspects of his identity remained his most valuable resources. Over the next few years, Omar worked part-time jobs for a variety of refugee oriented NGOs where he was hired specifically for his value as a proficient middleman between liberal, pro-refugee organisations and the people they purportedly helped. This became his biggest asset - he gave organisations access to the communities they targeted and his presence stamped a certain legitimacy to their activities. For much of the time that I knew Omar, this contradiction would repeatedly resurface. No matter how well he spoke or dressed like a Berliner, ultimately the only way he could articulate a case for his own economic value was by commoditizing aspects of his identity that he seemed desperate to erase. I've seen other Syrian refugees take this path, one fraught with this complex reconstitution of value and identity.

Indeed, the notion that neoliberalism has produced a rapid commoditization of ethnicity and experience (Comaroff & Comaroff, 2009) is well established. This is particularly emphasised in Berlin, and the desire of its hyper-mobile city elite to consume other-ness, and in the process constitute themselves as multicultural and cosmopolitan subjects. This is as true for the hundreds of new Syrian restaurants that have opened and closed since 2015, as it is for what I will tentatively call Berlin's refugee industry - a coming together of social work, NGOs, academic research grants, art exhibitions, films, and tourism, all centred around a consumption of refugee experience. People like Omar are the middlemen, the necessary lubricants in the functioning of this cosmopolitan engine.

\*\*\*

I'm meeting Omar at a Vietnamese restaurant in Prenzlauerberg. Despite my attempts to talk in Arabic, as always, Omar insists on responding in German. He begins by telling me that he had been fired from his job. He had been working at a fairly large German NGO that focuses on the cultural integration of newcomers. He tells me that a few weeks ago he told his team that he was going to be a father and would go on paternity leave. Given that he was still on his probationary period, and the fact that he had only told his teammates in person, Omar reckoned they cut their losses, knowing that he wouldn't be able to hold them accountable. "One person even informally told me that I could try again and apply once I'm back to work after my daughter is born. What else does that mean?"

The irony of this happening at an NGO explicitly aimed at assisting refugees doesn't elude either of us. After a good deal of frustrated commiserations, I finally ask him what he plans on doing next. "Honestly, I just don't want to do a job anymore. I want to start my own business. I have this idea, and I've been working on it for a while."

"You know, I've worked a lot in these kinds of NGOs. I'm always helping people organise events and concerts and stuff for refugees. But obviously, not that many refugees come. People always ask me why, and of course I know why. If you're just sitting at a camp somewhere, and you don't have any kind of employment, you won't go to a concert. I know because I was in a similar position. You can't go out to cultural events when you don't have money, but also when you feel like you're doing nothing, you don't even feel good when you go out. Well then the way to fix this… is to make sure they have jobs. So my idea is to get electric cycles - you know the ones with a big basket in the front? Yeah so to get those cycles and employ refugees to ride around Berlin selling groceries on them. You know, basic things, like water, chips, milk bread… that kind of stuff. And that'll also give Berliners the chance to help refugees daily - every time they buy from one of these cycles, they'll be directly supporting a refugee.

Omar goes on to elaborate, in great detail, the specifics of his plan. I realise that this isn't just some fanciful idea, and that he's worked out exactly what needs to happen - the permissions, the kinds of licences, the place to get the cycles made and even the initial capital required to make this work.

\*\*\*

Of particular importance here are  Omar's explanations for the functions and potential reasons for success in his new venture. What, if we were sitting in a slightly different context, might be called "the pitch". The fact that it starts with an explicit notion of how to get refugees more involved in Berlin's refugee industry is, I think, astute. Omar recognizes that all of the organisations that he's worked for rely on participation from refugees as refugees, and the fact that he starts with this in what is otherwise a fairly straightforward entrepreneurial plan indicates the extent to which he identifies the very real logic of the refugee industry in Berlin. The second "hook" of Omar's idea - the notion that this will allow Berliners to perform a kind of pro-refugee politics while buying from his employees - again hints at precisely this moral industry. Of course, what is surprising is not that someone might monetize moral activity, but that it is Omar, a refugee man himself, who is engaging in the commoditization of refugee-ness. It's the fact that he explicitly states, that it is precisely because he was in the same position as the refugees he hopes to employ, that he knows how to make them profitable. In one fell swoop, if successful, Omar will capitalise and commoditize aspects of his own identifications, while simultaneously taking advantage of the more general features of migrant economies and the desperation of those unable to find stable employment, located in the hierarchies of asylum seekers and refugees below him.

Though undoubtedly better off than others who worked in more precarious conditions, often, those of my interlocutors that found work in the "refugee industry", were frustrated by how they were reduced to becoming cultural brokers, regardless of their educational training or abilities. In her work on the employment of refugees in this sector, Sara de Jong argues that refugees are able to use "refugeeness as capital" (2019, p. 331) to open up previously unavailable  "windows of opportunity" in the labour market. Yet, as she observes, despite this work often being limited to highly educated refugees, "paradoxically refugees' qualifications from the country of origin are rarely acknowledged" ((2019, p. 334). As scholars have argued, it is important to recognize how "NGOs and smaller support organisations, wittingly or unwittingly, participate in the dequalification of migrant workers by offering a narrow range of roles and responsibilities" (Bird & Schmid, 2021, p. 15), and how nonprofit intermediaries can even more directly become involved in the disciplining and recruitment of labour in line with workfare transformations (Eick et al., 2003). Thus, though de Jong suggests that the recognition of "refugeeness" as capital acknowledges work in this sector as "skilled", for many it is precisely the conversion of "refugeeness" to capital that becomes a trap, reducing their economic

worth to their identifications as and with refugees. Even more crucially, as Omar's story illustrates, the notion of "refugeeness as capital" can also slip into "refugees as capital", which opens up the important distinction between those who can capitalise on refugeeness, and those who are turned into capital in the process. This is a tendency that permeates the nonprofit sector itself, with funding often being conditional on the participation of the "right kinds" of refugees, such that "NGOs and grassroots organisations come to owe their existence and economic sustainability to the continued production" of hierarchically related categories of deserving and undeserving refugees (Bird & Schmid, 2021, p. 17).

Omar belongs to the category of refugees that were given a holding environment within which to recuperate and "maintain a sense of continuity of self while living through discontinuous times" (Borneman, 2020, p. 40). His access to the full range of the welfare state, might indeed allow him to succeed in eventually retraining and transitioning away from the limited logic of the refugee industry. On the other hand, it seems often easier to name an injustice in the lower two segments, and one of the main challenges to people in Omar's position is the way Berlin's refugee industry provides him with a moral language of fidelity and charity that allows Omar to internalise a certain language and conception of selfhood. With this moral language also comes community, and a clearly defined position in relationship to non-refugees that, while overdetermined by his refugeeness, still allows Omar to have far more contact with non-refugees than any of his friends. In other words, escape in this segment, while technically possible, is linked to a loss of newly forged selfhood that might indeed prove an even more efficient anchor to the moral economy of Berlin's refugee industry. For those in Omar's position, the neo-liberal subjectivities demanded of them by the field of Asylum produce what Aihwa Ong (2003, p. 16) following the work of Foucault (1995)and Rose (1999) calls a "governance through freedom", through which "individuals also play a part in their own subjectification or self-making". These "entrepreneurial subjectivities are in part performed through a disarticulation of inequalities" (Scharff, 2016, p. 115). This disavowal of the role played by structural inequalities is central to understanding how refugees and migrants can themselves get invested in the endo-replication of the hierarchies they become enmeshed in.

**Productive Boundaries and the Production of co-ethnicity**

I've tried to describe earlier how the lowest segment of the migrant economy hierarchy is marked by *wage appropriation* and that the middle segment by a *policing and surveillance* system that involves migrant and refugee communities folding in on themselves. Those like Omar who find themselves at the top of the hierarchy of asylum seekers, are also not exempt from a logic marked by the commoditization of the cultural and ethnic aspects of selfhood in order to participate in the *refugee industry*. In doing so, I hope I have also provided clues on how these segments interact with one

another. If Omar's business were to succeed, the people he would make money off, would come from rungs below him. Khaled, placed at the middle rung, policed those from all three rungs - though most were probably from the intermediate and lowest ones. Conversely, those at the bottom of this hierarchy, like Haider, end up becoming cultural material and cheap labour for those above them, while constantly serving as reminders to those in the middle about the cost of losing the little security they've managed to cling to.

This process of differentiation and difference is at the core of what makes neo-liberal ethnic economies function, both to produce profit and value, and to perform the social and disciplinary role of bordering. While Haider uses the term "apne" (our own) to refer to his employers, it is the distinctions between them, both in terms of legal status, as well as in more cultural terms like caste, that define his experience of "co-ethnicity". As a security guard, Khaled was tasked with drawing on his own cultural prejudices and judgements to determine which of those "like him" were to be given access to spaces and which not. On the other hand, though he shared a language with them, Khaled's employers were Lebanese, and saw themselves as clearly apart from and above him. While Khaled uses the term "elarab mithluna" (Arabs like us) to describe his employers as well as the people he must "keep out" of the spaces he is tasked with guarding, it is clear that the identification in both directions is somewhat different. Further, Khaled's access to the social networks that fall within the "Arab" community in Berlin are markedly different from Omar's. And Omar himself was not part of the Urban Damascene community that often had the most resources and influence within the Syrian cohort of refugees.

The nuances of these divides, perceptible through dialects, last names and Habitus, are likely invisible to German bureaucracy. Yet they become operationalized in Germany through the unique intersections generated between the "moral economy of illegality" (Chauvin & Garcés-Mascareñas, 2012) on the one hand, and the cultural politics of difference within co-ethnic spaces on the other. Thus, it is important to keep in mind that the "co-ethnic" spaces are not apriori, but produced through the interactions of the identifications migrants bring with them, and those that are ascribed to them through German bureaucratic and economic infrastructures. As the anthropology of Fredrik Barth (1998) reminds us, often ethnic boundaries can exist not despite, but because of the market. Co-ethnicity is thus a consequence of the value of boundaries, not its cause. It defines how groups of proximate differences are delimited in ways that allow the subsequent boundaries to produce value in German economy and society.

Drawing on the work of those in the Black Radical tradition, scholars have used the term Racial Capitalism to articulate the relationship between the racialized differentiation of populations and capitalist and neoliberal economic practices (Gilmore, 2002; Melamed, 2015). More recently, such an

analysis has also been extended to refugees under the rubric of "relative surplus populations" (Bird & Schmid, 2021; Rajaram, 2018). Building on this body of work, we might think of the differentiation and segmentation of recently arrived refugees as part of "the disintegrating grind of partition and repartition through which racial capitalism perpetuates the means of its own valorization" (Gilmore et al., 2022, p. 30.51)". Simultaneously, as Jodi Melamed (2015, p. 79)argues, it is important to recognize the ideologies that are "key to racial capitalist processes of spatial and social differentiation that truncate relationality for capital accumulation". One such ideology, she suggests, is Multiculturalism, which "minoritizes, homogenizes, and constitutes groups as separate through single (or serial) axes of recognition (or oppression), repels accountability to ongoing settler colonialism, and uses identitarianism to obscure shifting differentials of power and unstable social relations". In other words, boundaries have dual functions - economic and ideological - and the interrelation of these two registers of boundary-work is central to understanding how they are produced, and how they function.

The question then becomes, how and when do ethnic boundaries become productive, and for whom? If the material goal of neoliberal policy shifts in the field of asylum has been to "activate" the labour potential of asylum seekers and refugees, then the ethnic economy is both its engine, as well as its alibi. The task of disciplining, bordering (and therefore ordering) new diversity has been subcontracted out - metaphorically and literally - to co-ethnic spaces. Thus, though unacknowledged (and intentionality is even harder to prove), co-ethnic networks are an essential part of the concrete assemblages that "integrate people and functions through modes of surveillance, regulation, punishment, and reward" (Ong, 2003, p. 10) and regulate arriving newcomers in Berlin.

The (dis)placement of migrant labour in ethnic economies works as a powerful alibi that disavows the role played by state policy in the differential and differentiated inclusion of new diversity in Germany. Migrants policing migrants, migrants expropriating wages from migrants, and yet other migrants working as the facilitators of the consumption and commodification of migrant experience and culture - all of these exploitative and disciplining relationships become the aesthetic scaffold that masks the relationship of newly arrived migrants to German economy and society. That is to say, if the goal is to make migrants productive, to police, surveil and discipline them, but also to enjoy the diversity and cosmopolitanism they bring, the aesthetic shift is the sleight of hand that makes it seem as though the violence of these relationships have nothing to do with the demands and desires of German society, but are instead the natural product of ethnic and cultural niches. The result is that they work as alibis for exclusion, for failed, conditional and limited incorporation.

In Berlin, this seems to have produced an autonomous engine of economic activity that contains some refugees within a sealed zone of refugee and migrant activity that produces value, and functions as a

project of social control, without challenging liberal Berliner sensibilities of injustice or outrage. Thus a famous Lebanese restaurant that employs new asylum seekers for far below the minimum wage, continues to be unrealistically cheap, and is roped in by local, progressive political organizations in a show of queer solidarity. Or a famously progressive bar uses refugee guards to keep refugees out, while the signs of "refugees are welcome here" or "refugees enter free" hang just out of reach.

**Conclusion: Vogelfrei Refugees**

Some might argue that refugees cannot expect both security in Germany and jobs of their choosing, and that this only represents an expectation of reciprocity from refugees. Germany has after all used large migration flows to bolster economic growth before. Since the 1960s, immigration in Germany "has been the "shock absorber of economic cycles (Konjunkturpuffer)", and immigration policies have complemented other economic policies" (Münz et al in Bauder, 2005, p. 103). So perhaps this is simply business as usual. However, as the field of asylum takes on the distinct characteristics of the political economy of migration, it does so under an entirely different ideological and moral framework. Georgina Ramsay (2020, p. 9) terms these innovations "humanitarian exploits", to refer "to situations in which the humanitarian imperative of saving lives becomes intersected with the economic imperative of making lives profitable: that is, to characterise situations of governance that are shaped by a dual imperative of protection and productivity." This subsumption of the economic migration model under the moral economy of Asylum has, in Germany, been referred to as a neo-liberal "post- Guest worker regime" (Buckel in Scherschel, 2016, p. 256). The shift from Guest workers to "Working Guests" poses serious questions about the possibility for moral and political claims about the nature and functions of labour, belonging and governance.

What is interesting is the moral reversal this represents. First, Turkish guest workers and their children were able to mobilise, organise, and demand rights in large part because no one could deny the role they played in rebuilding the country and bringing it back from the brink of economic collapse post WWII. Their "sustained and successful activism challenged the imposed category of "guest worker," switching the emphasis from guest to worker" (Miller, 2013, p. 226). These claims were an important part of the process that heralded fundamental changes to German society and its conception of itself (Koopmans & Statham, 1999). The idea, that asylum seekers must work off their obligation to the German state in exchange for security, threatens to erase this moral and political possibility, and this is ultimately what is represented by the absorption of asylum policy under the logic of immigration. It is a move that maintains hospitality as the gift that can never be completely repaid. Refugees can work like the migrants of the past, but their work might never reverse the moral hierarchy set up by the Asylum discourse. Further still, the coming together of a neoliberal discourse around deservingness and self-reliance under the bureaucratic rubric of "deservingness" disavows the

marginalising nature of the work refugees are employed in while simultaneously turning (some of) them and others in their co-ethnic networks into agents for the endo-replication of these hierarchies. There are a few more things that ought to be said here.

First, refugees, overwhelmingly, want to work. Indeed, as mentioned earlier, for decades, pro-refugee forces have campaigned for the right to work for refugees. Yet the current reforms produce a situation that expects refugees to quickly enter the labour market, often deprived of any symbolic, social, linguistic, or legal capital. Their degrees and certificates are devalued, they have few meaningful contacts when they arrive, and they are denied access to the rights that their "German" counterparts take for granted. They are, to borrow a Marxist phrasing, "Vogelfrei", or free as a bird. In German, Vogelfrei indicates a kind of ironic double freedom, that is to say, refugees are free to work, but they are also "free" from access to the kind of capital that German citizens are born with and accumulate throughout their lives.

Second, is the reminder that most refugees are not, in fact, people with asylum status, but tolerated foreigners. Yet despite this, and this is crucial to the argument presented here, the moral legitimacy claimed by Germany in Europe is based on the number of asylum seekers they let in - not the number of asylum applications they granted. The headlines that continue to emphasise the 1.2 million refugees that Germany opened its door to between 2015 and 2016 often fail to mention that, on average, Germany has granted asylum to less than 50% of those that have applied between 2015 and 2021. This may seem pedantic, but it is most certainly not, when the context in which the media politics of Germany's decision to open its doors becomes clear. This is the context in which Angela Merkel was named "Chancellor of the Free World" by time magazine, and even vocal left-wing detractors like Yanis Varufakis would claim that her actions were proof that "Europe's soul was not yet dead". The imaginative potential of where the 2015 moment placed Germany, as moral leaders of a new Europe relied, and continues to rely, on the hundreds of thousands of rejected asylum seekers living with precarious legal statuses in Germany.

Finally, I do not believe that this paper describes all, or even most, of the fields of economic activity that refugees are engaged in. Nor do I mean to imply the absence of solidarity. Khaled, at one point lost his apartment because he had helped a Syrian family get registered there, violating his rental contract. In turn, when struggling to find a new place, it was Omar who asked Khaled to move into his flat until he found something more permanent. Further, the support and help most of my interlocutors received from Germans in their social networks should not be understated. Nonetheless, it is important to recognize that "moments of autonomous solidarity do not in themselves undo the… violence of Racial Capitalism" (Bird & Schmid, 2021, p. 21). My point is not to suggest that intra and inter-ethnic forms of solidarity are absent, nor to suggest that these are not significant factors shaping

the everyday lives of refugees in Berlin. Instead, I wish to emphasise how conditions are created in which solidarity can frequently slip into maintaining, rather than overcoming, the boundaries on which capital depends.

What I have  outlined is a certain kind of logic that we have perhaps been reluctant to identify, although it may not always apply to other spheres of work inside and outside of co-ethnic spaces. Yet, while there may be spaces where this logic does not manifest itself, equally, I believe there are perhaps many others where it does, that I have either not been privy to, or not had my eyes open for. I wish to make clear that the analysis that I have presented displays a certain abstract specificity, a neat-ness, that anthropological work and indeed the complex experiences of my interlocutors, would not be able to sustain for long. Nonetheless, in doing so, what I hope to abstract out is a certain conceptual framework that might help contextualise, or ground, the very real messy-ness of anthropological investigations into the everyday lives of our interlocutors. If, therefore, my analysis seems on occasion heavy-handed, leaving little room for critique, I want to clarify that what I am outlining is not a totalizing framework for social action, but a heuristic one that hopes to recontextualize some emergent practices in the management of difference in Berlin and beyond.

At the time that I began writing this paper, Omar, from  Syria,  was unemployed and expecting a daughter with his German fiancé soon. He has big plans for a future that might free him from the narrow confines of his identity as a refugee, and it might well be that he will succeed. I lost touch with Haider, from Pakistan, several months before. His phone number no longer works. His wife was expecting their third child when I last met them. Khaled, from Egypt, spent months being unemployed, only to return to guard labour. Sensing that I was going to ask him if he had changed his mind about the nature of this work, he told me simply, "it's inhumane work, but at least inhumane work lets you be humane".

---

[i] While not everyone with a Duldung is a rejected asylum seeker, most are.

[ii] The history of the *Duldung* points precisely to this conclusion. The notion of temporary toleration gained popularity in Germany in the 90s in response to the conflict in Bosnia-Herzegovina and the war in Kosovo, though it has been used as a substitute for protection status since its inception in 1965 (Mitrić, 2013). The German state, and indeed other European states, made use of temporary toleration permits (which gave nation states much more flexibility) in-lieu of asylum under the 1951 United Nations Refugee Convention, with the focus on repatriation, rather than settlement (Gibney, 2000). The result, however, is that thousands of exiles continued to live with this status for decades after their arrival in Germany, seemingly with no end in sight to their "temporary" status.

[iii] Here I follow Etzold (2017) and Bauder (2005) in their use of Pierre  Bourdieu's (1977) notion of capital in examining migration and refugee contexts.

[iv] Further, for those with a Duldung, access to the labour market is not unconditional, and permission to work must be received from the local immigration authority. This is made further unlikely by the fact that applicants must already have a job contract before applying for permission, producing a chicken or egg situation where employers are unlikely to hire people who's right to work is itself uncertain.

[v] Anis Amri was a Tunisian Asylum seeker who was responsible for the 2016 attack on a Christmas market in Breitscheidplatz in Berlin that killed 12 people and injured 56 others.

**Acknowledgements**

I would like to  thank all my refugee interlocutors and friends without whom this and other parts of my project would not have been possible. The support and feedback I received for this article through Migration Politics residency, has been extraordinary, and I would like to thank Saskia Bonjour, Darshan Vigneswaran and, as well as my co-fellows, Ulrike Bialas and Salah Punathil for the effort and interest they showed in refining and polishing this paper. Thanks in particular to Evelyn  Ersanilli, whose multiple rounds of detailed feedback and generosity have been invaluable. In addition to their contributions, I wish to acknowledge the invaluable feedback I received on early drafts of this paper from my supervisor, John Borneman, as well as from  Franca van Hooren, and Maria Villares-Varela. I also want to thank the four anonymous reviewers, as well as the editor, for their detailed and thoughtful comments on this paper. Finally, I would also like to thank all those who gave feedback on a draft of this paper during the work for a work-in-progress session organised in Amsterdam.

**Funding Information:**

This project was made possible through the graduate financial support from Princeton University. A part of this project was supported by the DAAD (ref. number: 91734281).

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
