# Peer review of "Alibis of Exclusion: The role of ethnic economies in the differentiated inclusion of refugees in Berlin"

_Migration Politics_

## Round 1 · Referee Report · Simone Di Cecco · 2022-12-19

Strengths
1) as in my first report +
2) Important revisions have been made following referees reports : methodology, the text structure, some new theoretical reflexions and quotations.
Weaknesses
Many weakness stated in the first report have been adresse. There are still some minor changes about form and contents that can be done.
Report
The piece can be published in this journal, after it had been revised.
Requested changes
Formal changes :
- the abstract last sentence is clear and needs to be revisited
- there are often repetitions that can be cut (for instance "many hours" in paragraphe 2 page 3), and some problems with gaps between words
- "economic margins of the economy" (p.11) : sentence to be revisited
- the author keeps on using "***" in the text, but the usefulness of these marks is not clear
- some quoted authors are not in the list of references at the end (for instance Jodi Melamed)
- conclusions are too long and I think that the second-to-last paragraph is not necessary
Contents:
- I am still not sure about the relevance of using "refugees" for the population described here (the consequence of this choice is some unusual proposals like : "most refugees are not, in fact, people with asylum status")
- And still not sure about the relevance of using "ethnic" or "co-ethnic economy". Even if now the text is much more clear about choices made in this sens, these concepts are used for economic sectors not led by specific ethnic groups, and where specific ethnic groups are not the majority of the workforce (I'm thinking about security for instance). Why then talking about ethnicity for these sectors ? Because of the intermediation/low hierarchy ? Ethnicity plays an important role in recruitment processes as the author showed, but to me it does not allow to describe these sectors as ethnic or co-ethnic.
- In the title and in the text the author talks about alibis of "exclusion" : but it is in contrast with the theoretical choice of describing the refugees "differential inclusion" (Mezzadra, De Genova etc.). Why talking about exclusion?
- In the first part of the piece the author often uses the "camp" and "ethnic economy" nearly as synonyms (page 5 for instance), but the existing link between these two spaces/fields is not clear and not automatic : it needs to be explained.
Strengths
as in my first report --- plus the conceptual language, methodology (ethnographic approach, inductive analysis) are now much clearer --- the piece makes an original and topical contribution to Migration Politics
Weaknesses
I think the weaknesses stated in first report have been well addressed, especially some conceptual baggage around auto-ethnicity, a slightly essentialist concept of ethnicity within that, and a lack of considering how policy shapes migrants' livelihoods.
Report
I think the piece is a nice suit for the journal, after it had been revised.

---

## Round 1 · Author Response

1. On “auto-ethnic economies”
- Reviewers 1, 3 and 4 all expressed reservations about the validity of the term “auto ethnic economy”. Reviewer 1 points out that the words “auto” as well as “ethnic” don’t seem to work since the paper references practices where often other older settled migrants and/or German citizens are involved in these relationships. Further, Reviewer 1 suggested that my use of the term needs to be distinguished from classical concepts of the ethnic economy. Reviewer 3 felt the use of the notion of “economy” wasn’t substantiated well enough.
- I used the term “auto-ethnic economies” to try to capture the way economic activity, for my interlocutors, seemed to involve both a kind of isolation, as well as, as Reviewer 1 rightfully identifies, a kind of “endo-replication” of hierarchies. Upon reflection, I believe that introducing a new term here might be counterproductive for the clarity of the paper. The new manuscript now drops the term altogether, instead making the case for thinking of the functions and roles of ethnic economies under neoliberal transformations. The case the manuscript tries to make is that, under these transformations, the ethnic economy works both, to manage new diversity, as well as to function as an alibi for marginalisation that aesthetically or ideologically works to displace and disavow critique. I refer to these now under the rubric of “alibis of exclusion”, which function by masking a relationship between recently arrived migrants and the German economy and society as a relationship between migrants themselves” (p. 5). Thus, neoliberal ethnic economies, I contend, work as “both the engine and alibi of the German state’s attempt to produce value from recently arrived asylum seekers and refugees” (p. 5). To strengthen this claim, I have included references to the work of scholars working on middlemen and cultural brokers (Lindquist et al., 2012; Xiang, 2012), as well as to those working with theories and ideologies of racial capitalism (Gilmore, 2002; Jodi Melamed, 2015)
- Alongside this new focus, and given the observation from the Reviewers that I do not make much use of the “Guest Worker” paradigm until much later in the article, the manuscript now has a new title, “Alibis of Exclusion: The role of ethnic economies in the differentiated inclusion of refugees in Berlin”
- Reviewer 2 makes the case that I do not adequately demonstrate “how” changes in policy affect the labour relationships of my interlocutors. I have now included some more ethnographic material on p. 13-14 that addresses the way Haider saw the Duldung as preventing him from escaping exploitative co-ethnic relationships in two ways. First, by denying him the time needed to retrain and learn German given the timeline of deportation initiated by the rejection of his asylum plea, and second, because his legal status made it hard for him to articulate a case for his employability to other employers outside the ethnic economy. In a note (p. 12), I also add how the work permit for Duldung holders being conditional on them already having a job contract produces “a chicken or egg situation where employers are unlikely to hire people who’s right to work is itself uncertain”. Finally, I also add references to the work of other scholars who examine the relationship between uncertain legal statuses and employment (Lewis et al., 2015) as well as with migrants’ social worlds (Sigona, 2012).
2. The social construction of “co-ethnicity”
- All 4 Reviewers expressed concerns over the use of the co-ethnic analytic.
- Reviewer 2 pointed out that the usage of a certain rigid language was homogenising, particularly in regards to terms like “Arab”, “Asian” etc. I agree that this is a difficult line to tread. People are always far more than any one, or many, categories can describe. Nonetheless, people, and my interlocutors, also use certain ways to categorise themselves as parts of certain communities. Their use of these terms is, of course, dynamic. Someone that is “like you” in one context (eg. national identity), can become radically other in a different one (eg. religious identity). However, the fact that the use of categories of proximity are fluid is not, in my opinion, enough to justify not engaging with them at all. Taking into account Reviewer 2’s detailed comments, I have modified the language I use to indicate broader coalitions of categories where possible, (eg “their own linguistic, national or ethnic communities” on p. 4). Furthermore, the current manuscript emphasises how these categories were used by my interlocutors themselves - Haider’s use of “apne” (our own people), and Khaled’s use of “al arab mithluna” (Arabs like us). It does so not to absolve itself of the responsibility of considering the contradictions inherent in these notions - I take this on extensively (p. 14-15; p. 22-24) - but to emphasise the tragic and frustrating ways in which these economic relationships are experienced by my interlocutors.
- I have also tried to emphasise, and this is something Reviewer 1 also picks up on, the difference between “longer settled migrants” and “recently arrived refugees” within these economic relationships (p.5, p.6)
- Perhaps more crucially, as Reviewers 1 and 3 suggest, the manuscript now takes a more constructivist approach to “co-ethnicity” and ethnic boundaries more broadly. In a section titled “Productive Boundaries and the Production of co-ethnicity” (p. 22-24), the new manuscript engages more explicitly with the way co-ethnicity becomes a “consequence of the value of boundaries, not its cause”, arguing against a notion that thinks of it as “sui generis”. In line with Reviewer 1’s suggestions, I engage with the idea of Racial Capitalism to further this perspective, using the work of Gilmore (2002) and Melamed (2015), to talk about the way capital produces boundaries that become essential to its valorization.
3. Methodology
- All 4 reviewers wanted to see more detail on the research methodology and the question of my role and relationship in the field. The current manuscript provides some more detail on the research process.
- Reviewer 3 asked for clarity on whether the research was “fully inductive” or “grounded theory”. Though not explicitly using this terminology, the current manuscript now indicates that it was, in fact, inductive : “When I began, I did not intend to study the economic lives of my interlocutors specifically, but rather the broader dynamics of foreigner incorporation they were involved in, especially as initiated by interpersonal encounters in 2015/16. However, it quickly became clear to me that the field of labour was both unavoidable, and central to the nature of their engagements as and with members of Berlin’s social world.” (p. 4)
- The manuscript now also specifies the number of people I spoke with.
- Reviewer 3 also asked that I explain why I chose to tell these specific stories, and somewhat preempts my answer with their reference to Weber. The new manuscript now makes this more explicit on p. 6: “I chose to tell the stories of Omar Khaled and Haider because of how their experiences represented three distinct trajectories of labour incorporation and its relationship to co-ethnicity. Each had, as a result of their search for employment, unique relationships with other refugees, as well as with longer settled migrants of their own linguistic, national, or “ethnic” backgrounds. I began to think of their stories as representative - not necessarily in a statistical sense - of the dominant modes of engagement with the labour market for my interlocutors. Such a choice prioritises a kind of analytical specificity over complexity, eschewing stories that may not fit so clearly into the analysis that follows. Yet by doing so, I hope to articulate a possible way to think of the structural forces my interlocutors found themselves engaging with.”
- Reviewer’s 1, 2 and 4 all asked for more information about my relationship to my interlocutors, particularly in context of my volunteering work. The new manuscript now provides some more detail, indicating that I was a volunteer for 3 months (p. 3) where I made some initial contacts, before moving out of the camp over the course of the next 3 years of my fieldwork. On p. 12 I provide some more information on why I was asked to convince Haider to enrol in a vocational training course. The previous wording perhaps wrongly implied that this was my job at the camp, though as I now make clear, I worked mostly as a translator - I was the only Punjabi/Urdu speaker in the camp, and they believed that an Ausbildungsduldung would be the best way forward for him and therefore asked me to speak with Haider about it.
4. Structure
- Reviewers 1 and 4 both suggested I revert to a more traditional structure, with the background on the asylum policy coming all together in the introductory section. The new manuscript reflects this change, and I have moved up the interlude section on the Bleibeperspective to follow the section that details the neoliberal changes in asylum policy at the beginning of the paper.
- Reviewers 3 and 4 asked for clearer signposting at the end of the introduction. The new manuscript now has a detailed paragraph at the end that more explicitly outlines the structure of the paper, while also addressing what I believe to be the contributions of this paper, i.e. the relationship between the ethnic economy and neoliberal transformations of asylum policy.
5. Nomenclature, Precision
- Reviewer 2 rightly points out that the use of nomenclature - refugees, asylum seekers, newcomers etc. is not consistent, and occasionally erroneous and even homogenising. The new manuscript addresses this in a few ways. First, as I explain in a note at the beginning, “reflecting the self-identification of my interlocutors, I use the term refugee to refer to people that came to Berlin seeking asylum, irrespective of the legal statuses they were then granted by the German state. Where legal specificity is important, I use the terms “asylum seeker”, “rejected asylum seeker”, “tolerated foreigner”, “refugee with asylum status” etc.”
- Second, in places where the previous manuscript carelessly made claims that seemed to imply a truth for all refugees, the new manuscript takes care to be more specific. For instance in regards to private security, where the new manuscript specifies its popularity is among young, male asylum seekers and refugees (p. 16).
- Third, as mentioned earlier, the new manuscript clarifies the difference between recently arrived refugees and longer settled migrants and refugees within the sphere of work.
- Finally, the new manuscript has removed language that makes it seem like these legal statuses fully describe my interlocutors, indicating that these categories only define structural relationships with respect to the German state (p. 3)
- Reviewer 2 also points out some inaccuracies in the finer discussion on the Duldung and refugee status, as well as in the citing of certain references (for example the erroneous use of ‘differential exclusion’ instead of differential ‘inclusion’ in reference to Mezzadra and Neilsen’s work, as well as in the usage of the word “acceptance” of refugees with regards to the work of Drangsland and Schultz. The new manuscript corrects these mistakes, and I thank the Reviewer for their detailed attention and feedback on these finer points of the article.
6. Additional References
- In line with Reviewer 1’s suggestions the manuscript (p. 23-24) now engages with the notion of Racial Capitalism (Bird & Schmid, 2021; Gilmore, 2002; Gilmore et al., 2022; Jodi Melamed, 2015; Rajaram, 2018). Using the work of Bird and Schmid, as well as de Jong (2019), the manuscript (p. 21)now engages with the role of NGOs in maintaining what the manuscript refers to as “the refugee industry”. I take on de Jong’s notion of “refugeeness as capital” pointing to how “the notion of “refugeeness as capital” can also slip into “refugees as capital”, which opens up the important distinction between those who can capitalise on refugeeness, and those who are turned into capital in the process.”
- Reviewer 2 suggested engaging more extensively with existing work done on the Duldung. I thank the Reviewer for the suggestions, particularly Mitric’s dissertation, which I was completely unaware of. Though, for sake of brevity, I have not been able to fit in a more detailed discussion on the history of the category, in a note on the history of the Duldung, the manuscript now refers to the long history of the Duldung’s use as a category of protection, especially as it was operationalized in response to war and conflict in former Yugoslavia.
- Missing references have been added, both to substantiate claims made in the paper (for example on p. 7 that “asylum seekers faced serious hurdles in accessing the labour market, a policy meant to dissuade people thinking of applying for asylum in Germany (Hinger, 2020, p. 19))”, as well as to correct erroneously left out, or falsely entered, items in the bibliography.
7. Miscellaneous
- Reviewer 1 suggested that I engage with the counterfactual of solidarity, and asked if I hadn’t witnessed situations where the relationships described could be thought of under those terms. In the concluding section, the new manuscript now briefly engages with this. I show instances where solidarity is certainly present, but conclude, following Bird and Schmid, that “moments of autonomous solidarity do not in themselves undo the… violence of Racial Capitalism” (Bird & Schmid, 2021, p. 21)
- In line with Reviewer 1’s suggestions, the manuscript now has a note specifying that I use capital in the Bourdieusian sense.
- Reviewer 2 provided valuable feedback on how the demonstration of arguments might proceed from a “show don’t tell” approach. The current manuscript tries to reflect this approach, and includes ethnographic material before analysis, for example, Haider’s employers’ use of casteist slurs now precedes the analysis on p. 6 : Haider’s South Asian employers were able to draw on their reservoirs of social and cultural capital to read his situation in a way that he no doubt hoped employers from other ethnic groups might not be able. Their incessant use of casteist slurs and referring to him as from the “village” were not just meant to humiliate, but to also discipline and control through the performative reproduction of difference.
- Reviewer 4 felt that my analysis of the neoliberal transformation of asylum was too monocausal, particularly in regards to the right to work for those with Duldung status. While I agree that the picture the manuscript paints focuses on the role played by commercial lobbying interests, I now nuance this by providing both more detail on the importance of the right to work for those on a Duldung citing Tize (2021) on “permanent temporariness”. Nonetheless, I believe that the way in which these reforms took place emphasised the aims and goals of corporate lobbies rather than the needs and rights of tolerated foreigners themselves, such that not all those with the Duldung have experienced these transformations in the same way. Thus, the manuscript (p. 8) follows the conclusions of those like Theresa Schütze (2022, p. 16), who contend that “neoliberalization has not improved the situation for all persons with Duldung but has driven differentiation within the group and a rationale of ‘cherry picking’ by the state”.

---

## Round 2 · Author Response

I would sincerely like to thank both reviewers and the editor, for their continued efforts and comments on this manuscript. I have tried to incorporate as much of the new feedback as I felt possible, and below I detail some of the changes and my thoughts with regards to a few of the comments.

Reviewer 2 felt that the choice to use the term “refugee” was still under-justified in the manuscript. Accordingly, I have now moved my justification from and endnote to the main body of the text to clarify that I use the term because of the self-identification of my interlocutors, rather than as a result of the legal status they were ultimately able or unable to obtain. Further, I believe the fact that much of the bureaucratic apparatus that sorted my interlocutors in different sectors/segments of the economy was contained within the field of asylum. Thus while other migrants are likely subject to structures, my choice to focus on refugees is meant to emphasise the way an economization of the field of asylum was crucial to the story of how these new hierarchies emerge. To reflect this, I now also reference this reason in my justification in the main body of the text on p. 3.

Reviewer 2 also expressed reservations on my use of the term “ethnic” and “ethnic economy” particularly in the case of subcontracted labour. To clarify, while the overall security sector is certainly not an “ethnic economy” or an “enclave economy”, those of my interlocutors that were recruited through co-ethnic contacts worked, almost exclusively, in workforces made up of other refugees and migrants like themselves. I now clarify this on p. 18. There is some detail here that I have chosen to leave out - like the usage of permanent “white” employees in managerial positions. Yet, importantly, not only is the position of my interlocutors marginalised, as the Reviewer suggests, but it is relatively so - none of my Pakistani interlocutors were able to obtain these jobs, who lay outside these recruitment networks, and often settled for far more precarious labour. This is why I believe the use of the notion of “co-ethnicity” helps, because it defines not only a general marginality, but a dynamic and multiple process of boundary formation that is related to, but also exceeds, the market.

Reviewer 2 also pointed out a few grammatical and formatting errors which have been addressed in this manuscript. The new manuscript now also clarifies that my use of the “camp” is meant to be only allegorical, and though in the case of my interlocutors, there really was a literal return to the “camp”, I do not intend to make that the focus of my argument more broadly.

Finally the editor has asked for some clarity with regards to the formal/informal divide I allude to in my article. The new manuscript now reflects these revisions. On p. 13, I clarify that while Haider worked at a hotel, he worked “off the books”, and without a legal contract. Further on p. 18, I clarify how, while many of my interlocutors who worked in private security did have limited or temporary contracts, the were often, “illegally, made to work several additional hours off the book. This strategy of “organised informality” (Gooptu, 2013, p. 10) is central to understanding the functional role played by subcontractors who provide labour at lower costs while limiting legal liability for parent companies who conveniently look away from these practices.”

---

## Round 2 · List of Changes

The choice to use the term “refugee” is now justified in the main body of the text to clarify that the term is used based on the self-identification of interlocutors, rather than legal status.

The fact that much of the bureaucratic apparatus that sorted interlocutors in different sectors/segments of the economy was contained within the field of asylum is now referenced in the main body of the text on page 3.

The nature of the labour relations in private security is now clarified on page 18 to indicate that while the overall security sector is not an “ethnic economy” or an “enclave economy”, those interlocutors recruited through co-ethnic contacts worked, almost exclusively, in workforces made up of other refugees and migrants like themselves.

Grammatical and formatting errors have been addressed.

The use of the term “camp” is now clarified as allegorical and not the focus of the argument.

The formal/informal divide is now clarified on page 13 and 18 to indicate where interlocutors worked without legal contracts or "off the book" (Haider) , and where there were contracts but a situation of "organised informality", meant that much of the labour was nonetheless outside the scope delimited by the legal arrangements (Khaled and the Private security industry).

---

## Editorial Decision

unknown